# Effect of Mo and Cr on the Microstructure and Properties of Low-Alloy Wear-Resistant Steels

**DOI:** 10.3390/ma17102408

**Published:** 2024-05-17

**Authors:** Tian Xia, Yuxi Ma, Yunshuang Zhang, Jialiang Li, Hao Xu

**Affiliations:** School of Civil Engineering and Architecture, Wuhan Polytechnic University, Wuhan 430023, China; 20210812028@whpu.edu.cn (T.X.); 20220812019@whpu.edu.cn (Y.Z.); 12927@whpu.edu.cn (J.L.); xuhao33011@whpu.edu.cn (H.X.)

**Keywords:** low-alloy wear-resistant steel, cooling rate, martensite, alloying elements

## Abstract

Low-alloy wear-resistant steel often requires the addition of trace alloy elements to enhance its performance while also considering the cost-effectiveness of production. In order to comparatively analyze the strengthening mechanisms of Mo and Cr elements and further explore economically feasible production processes, we designed two types of low-alloy wear-resistant steels, based on C-Mn series wear-resistant steels, with individually added Mo and Cr elements, comparing and investigating the roles of the alloying elements Mo and Cr in low-alloy wear-resistant steels. Utilizing JMatPro software to calculate Continuous Cooling Transformation (CCT) curves, conducting thermal simulation quenching experiments using a Gleeble-3800 thermal simulator, and employing equipment such as a metallographic microscope, transmission electron microscope, and tensile testing machine, this study comparatively investigated the influence of Mo and Cr on the microstructural transformation and mechanical properties of low-alloy wear-resistant steels under different cooling rates. The results indicate that the addition of the Mo element in low-alloy wear-resistant steel can effectively suppress the transformation of ferrite and pearlite, reduce the martensitic transformation temperature, and lower the critical cooling rate for complete martensitic transformation, thereby promoting martensitic transformation. Adding Cr elements can reduce the austenite transformation zone, decrease the rate of austenite formation, and promote the occurrence of low-temperature phase transformation. Additionally, Mo has a better effect on improving the toughness of low-temperature impact, and Cr has a more significant improvement in strength and hardness. The critical cooling rates of C-Mn-Mo steel and C-Mn-Cr steel for complete martensitic transition are 13 °C/s and 24 °C/s, respectively. With the increase in the cooling rate, the martensitic tissues of the two experimental steels gradually refined, and the characteristics of the slats gradually appeared. In comparison, the C-Mn-Mo steel displays a higher dislocation density, accompanied by dislocation entanglement phenomena, and contains a small amount of residual austenite, while granular ε-carbides are clearly precipitated in the C-Mn-Cr steel. The C-Mn-Mo steel achieves its best performance at a cooling rate of 25 °C/s, whereas the C-Mn-Cr steel only needs to increase the cooling rate to 35 °C/s to attain a similar comprehensive performance to the C-Mn-Mo steel.

## 1. Introduction

As a high-performance wear-resistant material, low-alloy wear-resistant steel has the advantages of high strength and hardness, toughness, wear resistance, lower production costs, and convenient processing. It is widely used in construction, machinery, mining, electric power, chemical industry, and other fields [1,2,3,4,5,6,7]. However, the strength hardness and toughness of low-alloy wear-resistant steels have always been a pair of contradiction. The increase in strength and hardness usually leads to a decrease in toughness and plasticity [8,9]. Although this can be improved by reducing carbon content, trace alloying elements need to be added to ensure hardenability. Among them, Mo and Cr are considered important choices, effectively improving the matching of hardness and toughness in low-alloy wear-resistant steel and significantly enhancing comprehensive performance [10,11,12,13].

Extensive research has been conducted on the effects of Mo elements and Cr elements on the microstructure and properties of low-alloy wear-resistant steels. Rodriguez-Galeano et al. [14] studied the influence of Cr and Cr + Nb on the interphase precipitation and mechanical properties of V-Mo microalloyed steels. It was found that an addition of 0.5 wt% Cr to V-Mo microalloyed steel accelerated the transformation rate of ferrite, producing a higher volume fraction of ferrite. Importantly, the addition of Cr reduces the diameter of the interphase precipitates, making an important contribution to the yield strength. Xue et al. [15] studied the effect of hardenability on the microstructure and properties of low-alloy abrasion-resistant steel, comparing three different low-alloy wear-resistant steels with different Mo and Cr contents but the same heat treatment process. The results indicate that tempered martensite dominates throughout the thick plate, and steels with more Mo and Cr elements exhibit better strength and impact toughness; steels with a higher Mo content have a more uniform hardness distribution. Chu et al. [16] studied the influence of Cr content (0.3 wt.% and 1.0 wt.%) on the microstructure evolution and properties of low-alloy steel. The results show that the microstructures of low-alloy steel with 0.3 wt.% Cr are ferrite, granular bainite, martensite, and retained austenite, while no ferrite was observed in the microstructure of low-alloy steel with 1.0 wt.% Cr in the same process. Cr promotes the precipitation of (Nb, Ti)C in high-temperature austenite region, retards bainite transformation, and refines the grain size of low-alloy steel. Tang et al. [17] studied the influence of Cr/Mo on the corrosion and wear behavior of alloy steel. The wear test results showed that adding Cr/Mo improved the wear resistance of alloy steel, which can be attributed to the increased toughness and hardness gradient. Park et al. [18] studied the effects of three alloying elements, Ni, Cr, and Mo, on the mechanical properties of low-alloy steel. The research found that with increasing Cr and Mo element contents, the hardenability of low-alloy steel improved, leading to increased strength and fracture toughness. Cr and Mo primarily influenced the precipitation behavior of carbide phases, with the observation of Cr_23_C_6_, Cr_7_C_3_, and Mo_2_C carbide precipitations in the steel. Cr elements were found to inhibit the precipitation of cementite, reducing the size of carbides. Ayadi et al. [19] studied the microstructure and wear behavior of Cr, Mo, Nb alloy steels, and compared them with alloy steels with a lower Cr content. The results show that the additions of Cr, Mo, and Nb increased the rate of the formed martensite, which improved the hardness, wear resistance, and tribological behavior. Some studies also suggest that Mo can delay the growth of austenite grains, promoting grain refinement, and that both Mo and Cr can reduce the critical cooling rate for austenite transformation, facilitating bainitic transformation at low cooling rates and martensitic transformation at high cooling rates [20,21,22,23]. The Mo element can effectively increase the hardenability of steel and is one of the alloying elements commonly used to improve the hardenability of thick-specification wear-resistant steel plates, but the Mo element belongs to the precious alloying elements and scarce resources. Compared with the Mo element, the Cr element has a solid solution strengthening effect and lower price, which can effectively reduce the cost of steel production; however, but the role of the Cr element in enhancing hardenability is weak [24,25]. Overall, research on the influence of Mo and Cr elements on the microstructure and properties of low-alloy wear-resistant steel mainly focuses on different Mo contents, different Cr contents, and the combined addition of Mo and Cr elements. There are fewer comparative studies on the effects of separately adding Mo elements and Cr elements on the microstructure and properties of low-alloy wear-resistant steel at the same level. Additionally, considering that the cooling rate during quenching directly affects the phase transformation, microstructure, toughness, and hardness of steel, it is necessary to study the effects of separately adding Mo and Cr elements on the microstructure and properties of low-alloy wear-resistant steel under different cooling rates.

This study designed two types of low-alloy wear-resistant steels, namely, those with the Mo element solely added and those with the Cr element solely added, based on C-Mn series wear-resistant steels. In order to better elucidate the toughening mechanisms of Mo and Cr elements, a comparative study was conducted on the phase transformation, microstructure, and mechanical properties of the C-Mn-Mo steel and the C-Mn-Cr steel under different cooling rates to explore economically feasible production processes and provide new experimental data and theoretical validation for the composition, design, and production applications of low-alloy wear-resistant steels.

## 2. Materials and Methods

### 2.1. Experimental Materials

The chemical compositions of the two types of low-alloy wear-resistant steel are shown in Table 1. The experimental steel was melted in a 50 kg vacuum induction furnace and cast into Φ150 mm steel ingots. The rolling process was conducted in two stages, namely rough rolling and finish rolling, using a Φ800 mm experimental rolling mill. The detailed rolling process is as follows: In the rough rolling stage, the initial rolling temperature was set at 1080 °C, and rough rolling was performed from 1080 °C to 980 °C to obtain an 85 mm intermediate billet. Subsequently, in the finish rolling stage, precision rolling was performed at temperatures ranging from 950 °C to 910 °C, followed by finishing rolling at 850 °C, resulting in a steel plate with a final thickness of 20 mm. After rolling, it was cooled at a laminar cooling rate of 25 °C/s to below 200 °C before being air-cooled. The process parameters for the controlled rolling and controlled cooling of the experimental steel are shown in Table 2.

### 2.2. Experimental Procedures and Methods

The steel was tested for its thermal expansion curve using a Gleeble-3800 thermal simulation test machine (Poestenkill, NY, USA) (with an experimental temperature error of ±1 °C) to determine the phase transition temperature. The thermal expansion specimen of the experimental steel is shown in Figure 1. We conducted thermal simulation quenching experiments on experimental steel using a Gleeble-3800 thermal simulation test machine, under vacuum conditions, heating the experimental steel sample to 890 °C at a heating rate of 10 °C/s and holding it for 60 min. Then, different cooling rates were applied for thermal simulation quenching, where the cooling rates for the C-Mn-Mo steel were 20, 25, and 30 °C/s, respectively, and for the C-Mn-Cr steel, the cooling rates were 30, 35, and 40 °C/s, respectively. After quenching, tempering treatment was conducted in a heating furnace, with a tempering temperature of 250 °C and a holding time of 100 min. The specific heat treatment process is illustrated in Figure 2. The experimental steel Continuous Cooling Transformation (CCT) curve was plotted using JMatPro V13.0 software based on the chemical composition, quenching temperature, and holding time of the experimental steel.

Suitable samples were selected for mechanical grinding and polishing. They were then corroded using a 4% nitric acid alcohol solution for 15 s. The microstructure of the samples was observed using an OLYMPUS GX71 metallographic microscope (Tokyo, Japan). The tissue structure of the sample was further observed and analyzed using a JEM-2100F field emission transmission electron microscope (JEOL, Tokyo, Japan). The sample was prepared using the electrolytic double spray thinning method with a 15% alcohol solution of high-chloric acid, and the electrolysis time was 15 s.

Performance test samples of the same composition were taken from the same rolled steel plate. We prepared Φ10 mm tensile specimens and conducted tensile tests on an INSTRON 4204 electronic mechanical testing machine (Norwood, MA, USA) with a gauge length (L_0_) of 50 mm and a tensile rate set at 10^−3^/s. Three tensile specimens were tested to evaluate the tensile properties, and the average results were taken. We prepared standard impact specimens with a size of 10 mm × 10 mm × 55 mm for a Charpy-V notch test in the ZBC2452-3 pendulum impact testing machine (MTS, Eden Prairie, MN, USA) for impact performance experiments (pendulum blade radius of 2 mm). Using test temperatures of 20, 0, −20, −40 °C, we used three impact specimens to complete the impact performance test, recording the results of the test to obtain an average value. We measured the hardness using the BRIN200D-TL semi-automatic Brinell hardness tester (Foundrax, Somerset, UK) and took three measurements to obtain an average value. In order to study differences in the hardenability of the experimental steels, hardness measurements were conducted in the thickness direction. The sample size was 5 mm × 5 mm × 20 mm, the distance from the center of the indentation to the edge of the sample was 4 mm and the distance between two adjacent indentation centers was 3 mm, as shown in Figure 3. The temperature was 25 °C and a 1 mm diameter indenter was selected; the test force was 294.2 N, with a holding time of 15 s. Five measurement points were taken, with three measurements at each point, and an average value was calculated.

We conducted wear performance comparative tests on heat-treated specimens using a homemade friction and wear tester. This experimental apparatus is capable of simulating friction and wear conditions in actual mining operations in small- to medium-sized coal mines, as depicted in the schematic diagram in Figure 4. The upper test specimen was mounted into the fixture and placed on the spindle of the test machine. Subsequently, the lower test specimen was placed into the auxiliary chuck, checked for flatness, and then the screw was rotated to bring the upper and lower friction pairs close to each other for the wear test. Q345 steel was selected as the upper test specimen material for wear resistance, with a hardness of approximately 140 HB, while the lower test specimens were the C-Mn-Mo steel and the C-Mn-Cr steel. The experiment simulated an actual coal mining and transportation situation. We designed three sets of experimental parameters based on load and rotational speed data that were converted from the coal mining volume of the scraper conveyor in small- and medium-sized coal mines, in which the loads and rotational speeds were 100 N at 75 rpm/min, 150 N at 45 rpm/min, and 200 N at 20 rpm/min, and the friction time was 60 min, respectively. Before and after wear, ultrasonic cleaners were used to clean the samples, weigh them after drying, and calculate the amount of wear loss. An average result was taken twice for each set of tests.

## 3. Experimental Results

### 3.1. Phase Transition Temperature

The thermal expansion curve of the experimental steel is shown in Figure 5. The experimental steel samples expanded continuously on the Gleeble-3800 thermal simulation tester, exhibiting inflection points at phase transition temperatures. Using the tangent method, the austenite start transformation temperatures Ac1 for the C-Mn-Mo steel and the C-Mn-Cr steel were measured to be 700 °C and 715 °C, respectively, while the austenite end transformation temperatures Ac3 was measured to be 830 °C and 820 °C.

Using JMatPro software, the Ac3 temperature of the experimental steel was calculated based on its chemical composition. According to the empirical Formula (1) [26], the Ac3 temperatures of the two experimental steels were computed. The calculation results are shown in Table 3.
Ac3 = 925 − 219.4·√C − 7·Mn + 39·Si − 16·Ni + 13.5·Mo + 97·V(1)
where the symbol of the chemical element represents the mass fraction of the element (wt.%).

It can be seen from Table 3 that the Ac3 temperatures calculated by using the JMatPro software are basically consistent with those measured by the dilatometric method, which proves the accuracy of the simulation calculation performed by using JMatPro software. The quenching heating temperature is generally about 50 °C above the Ac3 temperature, so the heating temperature of the experimental steel is Ac3 + 50 °C [27,28,29], and the quenching temperature was determined to be 890 °C, holding for 60 min to ensure complete austenitization.

### 3.2. CCT Curve and Phase Transition Laws

The CCT curves of the two experimental steels were simulated using JMatPro software, as shown in Figure 6. These curves reflect the influence of phase transformation temperatures, the extent of phase transformation, and the microstructure of the experimental steels. It can be observed that for the C-Mn-Mo steel, the phase transformation temperatures are as follows: Ac1 is 705 °C, Ac3 is 828 °C, M_s_ is 410 °C, and M_f_ is 302 °C. For the C-Mn-Cr steel, the phase transformation temperatures are as follows: Ac1 is 719 °C, Ac3 is 819 °C, M_s_ is 407 °C, and M_f_ is 297 °C. At a cooling rate of 0.1 °C/s, the undercooled austenite transformation microstructure of the C-Mn-Mo steel and the C-Mn-Cr steel mainly consists of ferrite (F), pearlite (P), and bainite (B) microstructures. At a cooling rate of 1 °C/s, the pearlite gradually disappears in the C-Mn-Mo steel, and the undercooled austenite transforms into ferrite and bainite microstructures. In the C-Mn-Cr steel, the undercooled austenite transforms into ferrite, bainite, and a small amount of pearlite microstructures. When the cooling rate increases to 10 °C/s, the microstructure of both test steels consists of martensite, ferrite, and bainite. When the cooling rate exceeds 13 °C/s, the ferrite and bainite disappear in the C-Mn-Mo steel, and it enters the fully martensitic transformation region. When the cooling rate exceeds 24 °C/s, the ferrite and bainite in the C-Mn-Cr steel disappear, and it enters the fully martensitic transformation region. Comparative studies on the effects of Mo and Cr on the microstructural transformation and mechanical properties of low-alloy wear-resistant steels at different cooling rates were conducted through thermal simulation experiments. For the C-Mn-Mo steel, cooling rates were set to 20, 25, and 30 °C/s, while for the C-Mn-Cr steel, cooling rates were set to 30, 35, and 40 °C/s.

### 3.3. Microstructure Analysis

The microstructures of hot-rolled the C-Mn-Mo steel and the C-Mn-Cr steel are shown in Figure 7. The observation results indicate that the microstructures of the two experimental steels in the hot-rolled state are very similar, consisting of ferrite (white) and pearlite (dark black), with ferrite exhibiting a polygonal block distribution.

Figure 8 illustrates the microstructure of the two experimental steels cooled to room temperature at different cooling rates. The observation results indicate that the microstructures of the two experimental steels obtained under different cooling rates are mainly martensite (M), with a small amount of ferrite (F) and bainite (B), and no pearlite structure. As shown in Figure 8a, at a cooling rate of 20 °C/s, the microstructure of the C-Mn-Mo steel consists of martensite, bainite, and a small amount of ferrite. As depicted in Figure 8b,c, at cooling rates of 25 °C/s and 30 °C/s, the ferrite essentially disappears, and the microstructure of the C-Mn-Mo steel is predominantly martensitic with minor amounts of bainite. The martensitic structure is refined, evenly distributed, and exhibits distinct plate-like features. Under a cooling rate of 30 °C/s, the martensitic structure of the C-Mn-Cr steel contains a small amount of bainite and ferrite, as shown in Figure 8d. When the cooling rate increases to 35 °C/s, the ferrite in the structure disappears, and the structure is composed of martensite and a small amount of bainite, as shown in Figure 8e. This is because during the continuous cooling process, carbon atoms precipitate from the ferrite and enter the austenite to form carbon-rich austenite. The carbon-rich austenite further transforms to form martensite and bainite. As shown in Figure 8f, when the cooling rate is 40 °C/s, the microstructure of the C-Mn-Cr steel remains basically unchanged. Overall, the most suitable cooling rates for the C-Mn-Mo steel and the C-Mn-Cr steel are 25 °C/s and 35 °C/s, respectively. Excessively fast cooling rates may lead to excessive hardness and increased brittleness of the martensite structure. Changes in cooling rates did not result in the experimental steel acquiring other microstructure types (such as bainitic structure, austenite/bainite structure, etc.) of low-alloy wear-resistant steel. Therefore, it can be inferred that the differences in the mechanical properties of the experimental steel are primarily caused by the martensitic structure formed at different cooling rates.

To further analyze the microstructure of the C-Mn-Mo steel at a cooling rate of 25 °C/s and of the C-Mn-Cr steel at a cooling rate of 35 °C/s, transmission electron microscopy was used to observe their respective microstructures. The results are shown in Figure 9. It can be seen that the microstructures of the experimental steels were mainly lath martensite with a martensitic volume fraction above 98%. From Figure 9a,b, it can be observed that both the C-Mn-Mo steel and the C-Mn-Cr steel exhibit a lath-like martensite structure in the quenched condition. The lath features of the C-Mn-Mo steel are very pronounced, with a uniform distribution of laths, while in the C-Mn-Cr steel, fine ε-carbides can be seen. The transmission electron microscopy microstructures of the two test steels after low-temperature tempering at 250 °C are shown in Figure 9c,d. It can be observed that the martensite boundaries in the C-Mn-Mo steel are clear, and a high density of dislocation tangles are observed, along with a small amount of residual austenite. In the tempered microstructure of the C-Mn-Cr steel, there are numerous lath martensites, and granular ε-carbides precipitating within and around the martensite laths can be clearly observed. Compared to the C-Mn-Mo steel, the C-Mn-Cr steel has a lower content of residual austenite and a higher amount of ε-carbide precipitation.

### 3.4. Mechanical Properties of Experimental Steel

The mechanical properties of the C-Mn-Mo steel and the C-Mn-Cr steel at different cooling rates can be found in Table 4. According to the table, with the increase in cooling rate, the strength and hardness of the C-Mn-Mo steel increase, while the elongation decreases. The impact toughness initially increases and then decreases. When the cooling rate increases to 25 °C/s, the yield strength reaches 1100 MPa, the tensile strength reaches 1295 MPa, the elongation after fracture is 12.5%, the hardness reaches 390 HB, and the −40 °C impact toughness is 34 J. When the cooling rate rises to 30 °C/s, the hardness of the C-Mn-Mo steel continues to increase while the impact toughness decreases. With the increase in cooling rate, the strength and hardness of the C-Mn-Cr steel increase, while the elongation decreases. The impact toughness first increases and then decreases. When the cooling rate reaches 35 °C/s, the yield strength of the C-Mn-Cr steel reaches 1112 MPa, tensile strength reaches 1310 MPa, hardness reaches 395 HB, elongation is 11.3%, and the impact toughness at −40 °C reaches a maximum of 29 J. When the cooling rate reaches 40 °C/s, the low temperature impact toughness of the C-Mn-Cr steel decreases. Overall, the C-Mn-Mo steel and the C-Mn-Cr steel achieve the best matching of hardness and toughness at cooling rates of 25 °C/s and 35 °C/s, respectively. At this point, the tensile strength of the C-Mn-Cr steel is 15 MPa higher than that of the C-Mn-Mo steel, and its hardness is 8 HB higher than that of the C-Mn-Mo steel. The impact toughness of the C-Mn-Mo steel at −40 °C is 5 J higher than that of the C-Mn-Cr steel. Combining the microstructures of the experimental steels, it is not difficult to see that with the increase in cooling rate, the mechanical properties and microstructure of the C-Mn-Cr steel show similar effects to the C-Mn-Mo steel.

A detailed analysis was conducted on the mechanical properties of the C-Mn-Mo steel under a cooling rate of 25 °C/s and on the C-Mn-Cr steel under a cooling rate of 35 °C/s, including stress–strain curves, impact toughness at different temperatures, and hardness variation in the thickness direction. Figure 10 shows the tensile stress–strain curves of the two experimental steels. From the graph, it is clear that due to the influence of the martensitic matrix, both have elongation between 11 and 13%. Compared with the C-Mn-Mo steel, the C-Mn-Cr steel has a slightly higher strength and a slightly lower elongation.

Figure 11 shows the impact test results of the two tested steels at different temperatures. As the temperature increases, the impact toughness of both steels gradually increases. Specifically, the impact toughness values of the C-Mn-Mo steel at −40, −20, 0, and 20 °C are 34, 45, 51, and 69 J, respectively, while the impact toughness values of the C-Mn-Cr steel at −40, −20, 0, and 20 °C are 29, 43, 50, and 70 J, respectively. In low-temperature environments, both the tested steels demonstrate good impact performance, with the C-Mn-Mo steel exhibiting higher impact toughness than the C-Mn-Cr steel. However, as the temperature increases, this difference gradually diminishes. At 20 °C, the impact toughness of the C-Mn-Cr steel even surpasses that of the C-Mn-Mo steel.

Hardness is an important mechanical property parameter in low-alloy wear-resistant steel and is often used to evaluate its wear resistance [30]. The hardness test results along the thickness direction of the experimental steels are shown in Figure 12. It can be observed that the hardness of the experimental steels gradually decreases from the edge to the center. The hardness at the center of the C-Mn-Mo steel was slightly lower than that at the edge, with an average hardness of 385 HB close to the surface hardness and a minimum value of 374 HB. The average hardness of the C-Mn-Cr steel is 382 HB, with a center hardness that is lower than at the edge and a minimum value of 363 HB.

### 3.5. Wear Performance

The wear performance of the two experimental steels was tested using a homemade friction and wear test machine, comparing the differences in wear performance under different loading conditions. Table 5 presents the weight loss of the two experimental steels under different conditions during the wear process. It can be seen that as the load increases, the amount of wear of the two experimental steels increases and the amount of wear under the action of 200 N increases significantly. At a load of 100 N, the wear of the C-Mn-Cr steel is less than that of the C-Mn-Mo steel. However, at loads of 150 N and 200 N, the wear of the C-Mn-Cr steel exceeds that of the C-Mn-Mo steel.

## 4. Analysis and Discussion

According to the CCT curves of the experimental steels shown in Figure 6, it can be observed that the phase transition temperatures of the C-Mn-Mo steel and the C-Mn-Cr steel are very close to each other, with Ac3 values of 828 °C and 820 °C, and M_s_ values of 410 °C and 407 °C, respectively. However, there is a significant difference in the critical cooling rates for martensitic transformation, which are 13 °C/s and 24 °C/s, respectively. For low-alloy wear-resistant steels, the addition of Mo and Cr can improve hardenability, refine austenite grains, and decrease the temperature at which austenite transforms into martensite [12,16,25]. Thus, it can be seen that the addition of Mo effectively suppresses the transformation of ferrite, reducing the critical cooling rate for martensitic transformation and promoting martensitic transformation. The addition of Cr lowers the temperature at which the austenite phase transformation ends, narrowing the range of the austenite phase.

This study investigates the impact of Mo and Cr elements on the microstructure and properties of low-alloy wear-resistant steel under varying cooling rates. Figure 8 shows that when the cooling rate approaches the critical cooling rate for complete martensitic transformation, a small amount of ferrite and pearlite appear in the test steel. As the cooling rate increases, the ferrite and pearlite gradually disappear. This is because the increase in cooling rate enhances the stability of the undercooled austenite transformation and the nucleation driving force for ferritic transformation, thereby promoting the occurrence of martensitic transformation [31,32]. Combining the results shown in Figure 6 and Figure 8, it can be seen that the phase transformation behavior of the experimental steel at different cooling rates basically conforms to the CCT curve. The cooling rate plays a crucial role in the microstructural evolution of the C-Mn-Mo steel and the C-Mn-Cr steel. When the experimental steel is cooled below the M_s_ temperature, the supercooled austenite undergoes martensitic transformation. Due to the instability of martensite, it is necessary to control the cooling rate within the range of 20 °C/s or higher to avoid carbon diffusion in oversaturated martensite. The addition of the Mo element increases the amount of martensite transformation during the cooling process. This is because Mo atoms diffuse slowly at the austenite grain boundaries, slowing down the transformation from austenite to ferrite and hindering ferrite formation. Cooling rates above 25 °C/s have little effect on the structure of the C-Mn-Mo steel, while they significantly promote martensite transformation in the C-Mn-Cr steel. Therefore, the optimal cooling rate range for the C-Mn-Cr steel is different from that of the C-Mn-Mo steel. By adjusting the cooling rate it was found that at a cooling rate of 35 °C/s, the C-Mn-Cr steel obtained a similar microstructure to the C-Mn-Mo steel, as shown in Figure 8b,e. The plate-like characteristics of martensite were obvious, with fine and evenly distributed sizes, ensuring that the experimental steel had higher strength and hardness. Further examination was carried out on the microstructures of the C-Mn-Mo steel cooled at 25 °C/s and the C-Mn-Cr steel cooled at 35 °C/s. As per the observations under transmission electron microscopy depicted in Figure 9, low-temperature tempering produced a tempered martensite structure, which effectively reduced quenching stress [33,34,35]. After tempering at 250 °C, the martensite lath characteristics of the C-Mn-Mo steel become obvious, with a small amount of retained austenite and the phenomenon of high-density dislocation tangles observed, which restricts the slip of mobile dislocations, thereby playing a significant strengthening role. The residual austenitic film distributed between the martensitic slats can weaken the influence of internal stress concentration due to the high density of the slats and prevent the generation of cracks between the martensitic slats. This is a key factor to ensuring that steel has good toughness [36,37]. Unlike the Mo element, the inhibition of dislocation movement of the Cr element reduces dislocation density and dislocation entanglement. There are more ε-carbides precipitated in the C-Mn-Cr steel, which plays a role in pinning dislocations and promotes the strength and hardness of the experimental steel to be improved.

Enhancing the wear resistance of steel has always been an important goal in the research of low-alloy wear-resistant steels, and the wear resistance of steel mainly depends on its hardness and toughness. Increasing hardness can reduce the surface wear of steel and enhance its ability to resist wear. Good toughness enables steel to have excellent resistance to impact and crack propagation, preventing the detachment of particles on the contact surface during wear and thereby slowing down the wear process [38,39,40]. According to the mechanical performance results in Table 4, the strength and hardness of the experimental steel increase with the increase in cooling rate, while the low-temperature impact toughness decreases. A comparison between two experimental steels shows that the low-temperature impact toughness of the C-Mn-Mo steel is slightly superior due to its finer martensite lath structure and the presence of residual austenite in small amounts. However, the strength and hardness of the C-Mn-Cr steel increase more significantly with varying cooling rates, as the Cr element enhances the steel’s strength and hardness through second-phase strengthening. Therefore, Mo enhances low-temperature impact toughness more effectively than Cr, while Cr has a more noticeable effect on strength and hardness compared to Mo. As shown in Figure 11, the impact toughness of the C-Mn-Mo steel and the C-Mn-Cr steel are almost the same at −20, 0, and 20 °C, with only slight differences at −40 °C, which may be attributed to factors such as the microstructure, grain size, and inclusions of the experimental steel [41,42]. The C-Mn-Cr steel has a high precipitation strengthening effect, which adversely affects the toughness, so the impact work at −40 °C is not as high as for the C-Mn-Mo steel. The hardness test results along the thickness direction of the experimental steel in Figure 12 show that the average hardness values of the C-Mn-Mo steel and the C-Mn-Cr steel along the thickness direction are close at 385 HB and 382 HB, respectively. A comparison reveals that the edge hardness of the C-Mn-Cr steel is higher than that of the C-Mn-Mo steel, while the core hardness is lower than that of the C-Mn-Mo steel, indicating that the improvement effect of Mo on the hardenability of low-alloy wear-resistant steel is greater than that of Cr. However, the Cr element plays a good role in solid solution strengthening and precipitation strengthening, thereby increasing the surface hardness. Overall, the C-Mn-Mo steel cooled at a rate of 25 °C/s and the C-Mn-Cr steel cooled at a rate of 35 °C/s, exhibiting similar mechanical properties. Through wear experiments comparing the difference in the wear performances of the experimental steels with the addition of Mo alone and Cr alone, it can be seen that under the same friction time conditions, the greater the load, the more significant the increase in weight loss caused by friction wear, as shown in Table 5. Compared to the C-Mn-Mo steel, the C-Mn-Cr steel exhibits relatively higher strength and surface hardness. Under low loads (100 N), the weight loss caused by frictional wear is smaller. However, under high loads (150 N, 200 N), due to the relatively uniform hardness distribution of the C-Mn-Mo steel and its superior toughness compared to the C-Mn-Cr steel, the weight loss caused by frictional wear is slightly smaller than that of the C-Mn-Cr steel.

## 5. Conclusions

The effects of 0.2 wt.% Mo and 0.5 wt.% Cr on the microstructure and properties of low-alloy wear-resistant steel were studied, and the following conclusions were reached:(1)The transformation temperatures of the C-Mn-Mo steel and the C-Mn-Cr steel are close, with the Mo element exerting a greater promotion effect than the Cr element on the martensitic transformation rate. Among them, the Ac3 values are 828 °C and 819 °C, respectively, and the M_s_ values are 410 °C and 407 °C, respectively. The critical cooling rates for complete martensitic transformation are 13 °C/s and 24 °C/s, respectively.(2)The Mo element effectively inhibits the transformation of ferrite and pearlite, while the Cr element reduces the austenite transformation temperature and narrows the temperature range of austenite transformation. The Mo and Cr alloying elements can both effectively enhance the mechanical properties of experimental steel. The addition of the Mo element has a better effect on improving impact toughness and elongation at low temperatures, while the addition of the Cr element leads to more significant enhancements in strength and hardness.(3)With the increase in cooling rate, the martensitic structures of both the experimental steels gradually refined, and the lath characteristics gradually appear. The C-Mn-Mo steel exhibits a higher dislocation density, showing dislocation entanglement phenomenon, and contains a small amount of residual austenite, while in the C-Mn-Cr steel, the precipitation of granular ε-carbides is evident.(4)The C-Mn-Mo steel and the C-Mn-Cr steel achieve martensitic structures and optimal mechanical properties at cooling rates of 25 °C/s and 35 °C/s, respectively. Their tensile strengths reach 1295 MPa and 1310 MPa, respectively, with elongations of 12.5% and 11.3%. The impact toughness values at −40 °C are 34 J and 29 J, respectively, while the surface hardness values are 390 HB and 398 HB, respectively, with average hardness values along the thickness direction of 385 HB and 382 HB, respectively.

## Figures and Tables

**Figure 1 materials-17-02408-f001:**
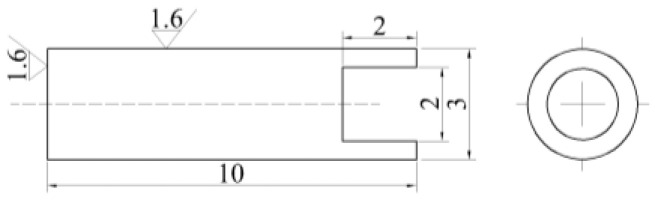
Thermal expansion specimen of the test steel (unit: mm).

**Figure 2 materials-17-02408-f002:**
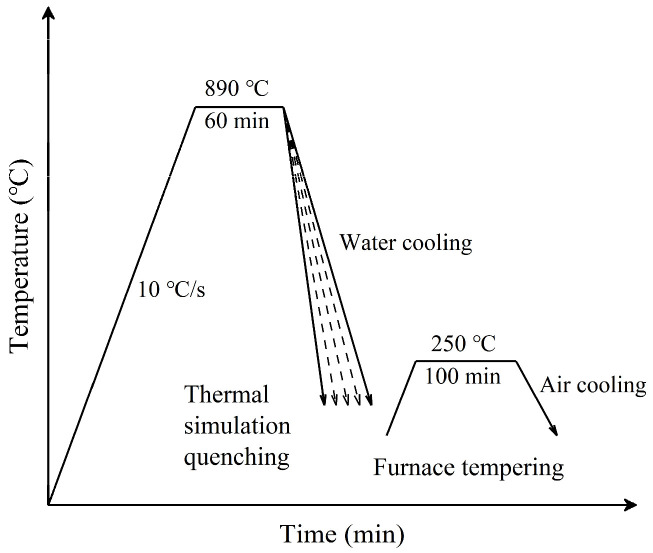
Schematic diagram of hot simulation quenching and furnace tempering.

**Figure 3 materials-17-02408-f003:**
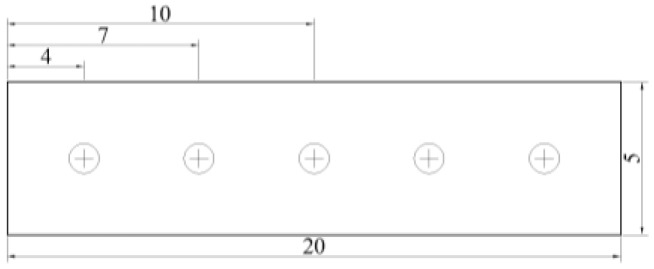
Hardness sample of experimental steel along the thickness direction (unit: mm).

**Figure 4 materials-17-02408-f004:**
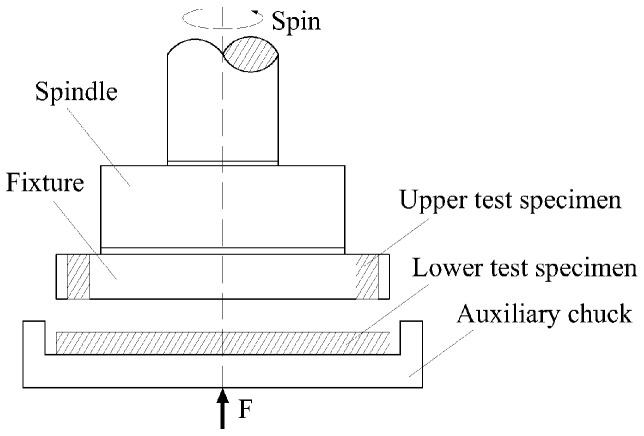
Schematic diagram of friction and wear experiment.

**Figure 5 materials-17-02408-f005:**
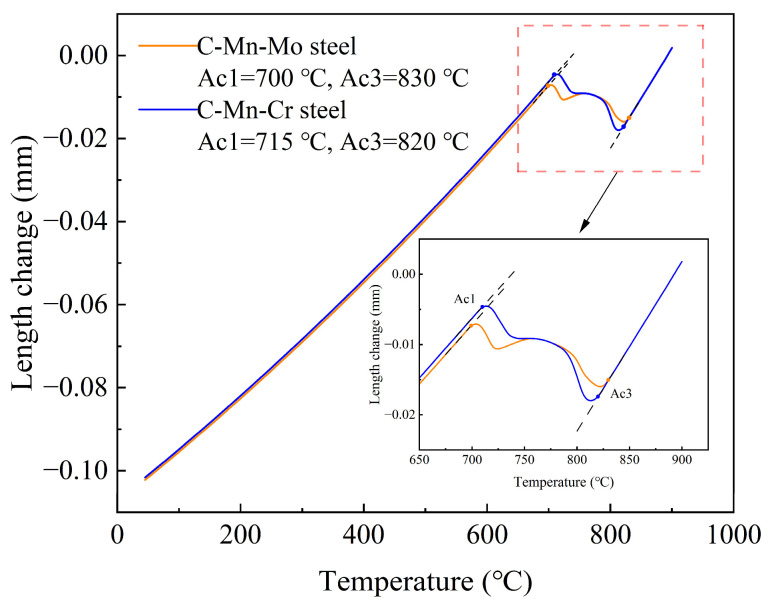
Thermal expansion curve of test steels.

**Figure 6 materials-17-02408-f006:**
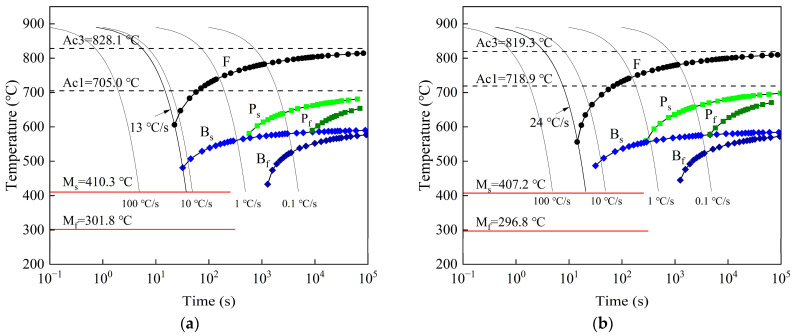
Experimental steel CCT curve. (**a**) C-Mn-Mo steel; (**b**) C-Mn-Cr steel.

**Figure 7 materials-17-02408-f007:**
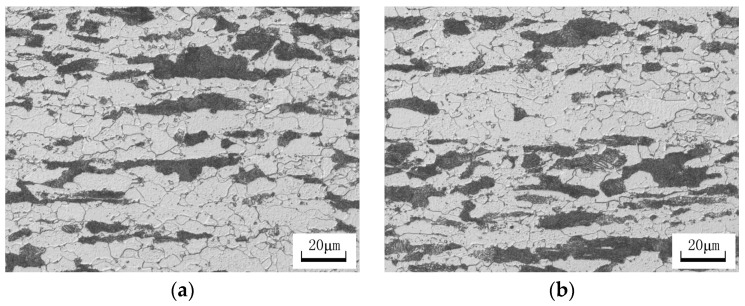
Hot rolled microstructure of experimental steel. (**a**) C-Mn-Mo steel; (**b**) C-Mn-Cr steel.

**Figure 8 materials-17-02408-f008:**
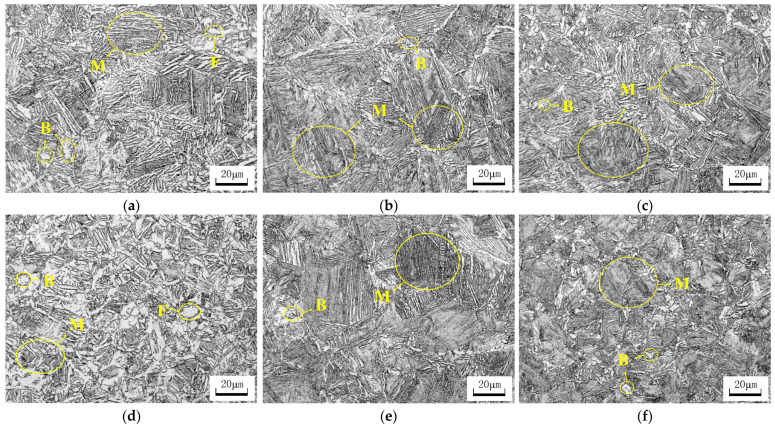
Microstructure of experimental steel at different cooling rates. (**a**) 20 °C/s C-Mn-Mo steel; (**b**) 25 °C/s C-Mn-Mo steel; (**c**) 30 °C/s C-Mn-Mo steel; (**d**) 30 °C/s C-Mn-Cr steel; (**e**) 35 °C/s C-Mn-Cr steel; (**f**) 40 °C/s C-Mn-Cr steel.

**Figure 9 materials-17-02408-f009:**
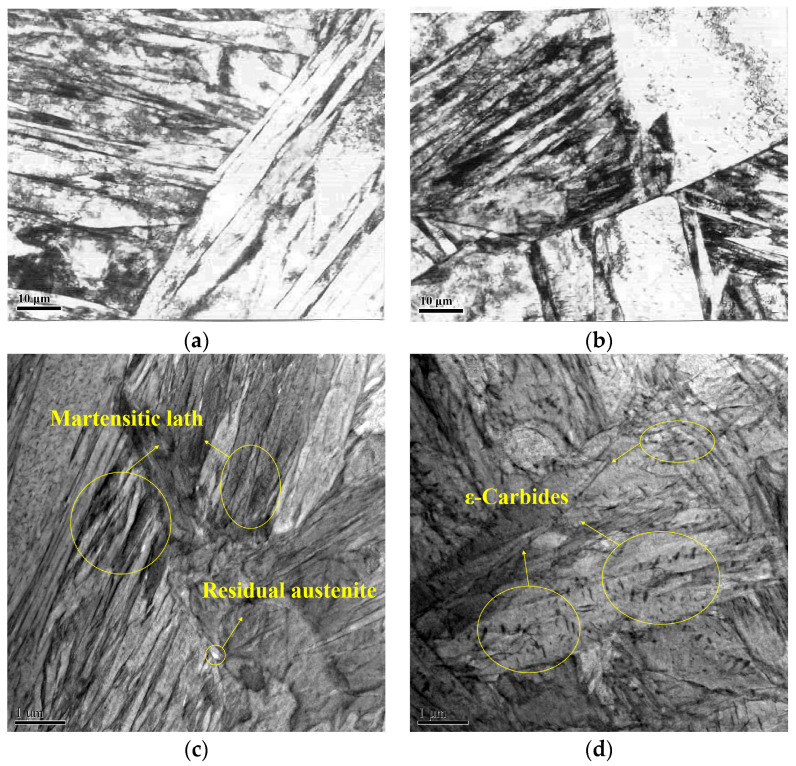
TEM organization of the test steels. (**a**) Quenching microstructure of C-Mn-Mo steel; (**b**) quenching microstructure of C-Mn-Cr steel; (**c**) tempered microstructure of C-Mn-Mo steel; (**d**) tempered microstructure of C-Mn-Cr steel.

**Figure 10 materials-17-02408-f010:**
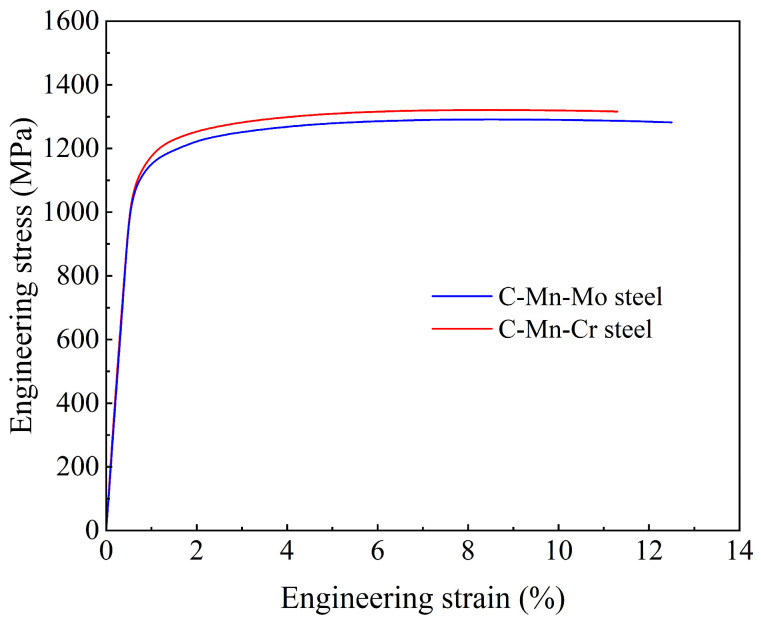
Experimental steel stress–strain curve.

**Figure 11 materials-17-02408-f011:**
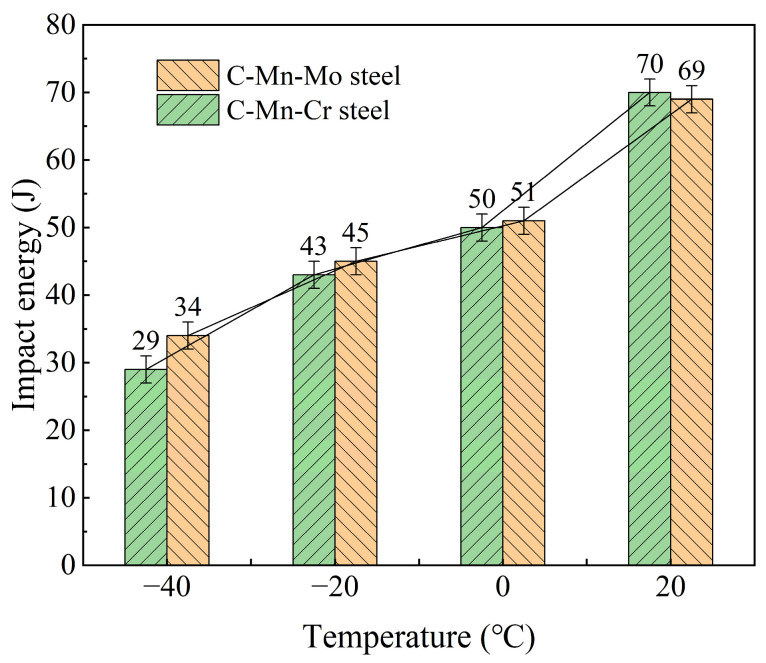
Impact energy of experimental steel at different temperatures.

**Figure 12 materials-17-02408-f012:**
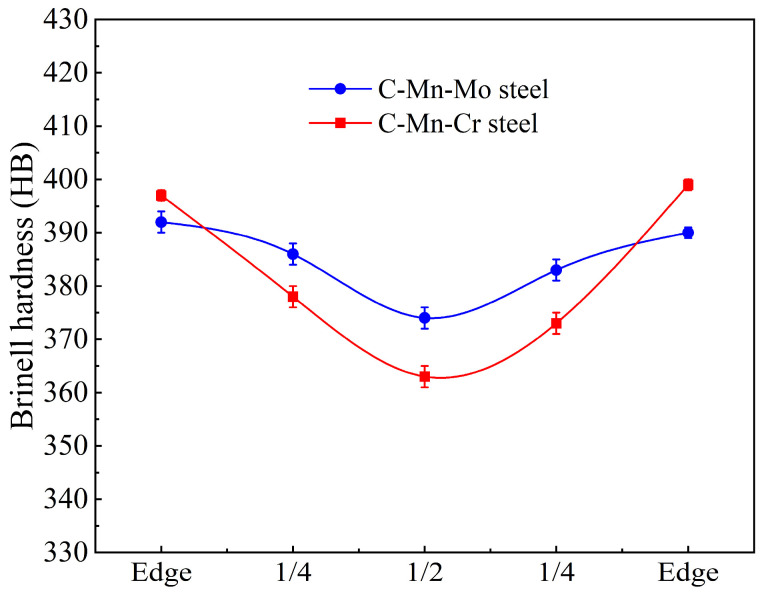
Hardness changes in the thickness direction of experimental steel.

**Table 1 materials-17-02408-t001:** Chemical composition of the tested steel (mass fraction, wt.%).

Material	C	Si	Mn	P	S	Mo	Cr	Ti	Nb	B	Fe
C-Mn-Mo steel	0.172	0.36	1.36	0.015	0.002	0.20	-	0.016	0.022	0.0013	Rest.
C-Mn-Cr steel	0.174	0.37	1.33	0.015	0.002	-	0.51	0.017	0.023	0.0012	Rest.

**Table 2 materials-17-02408-t002:** Process parameters of controlled rolling and controlled cooling.

Material	Steel Plate Thickness, mm	Heating Temperature, °C	Rough Rolling Stage Temperature, °C	Intermediate Billet Thickness, mm	Finish Rolling Start Temperature, °C	Finishing Temperature, °C	Cooling Rate, (°C/s)
C-Mn-Mo steel	20.1	1080	1080–980	85.3	945	855	25
C-Mn-Cr steel	20.0	1080	1080–980	84.5	949	853	25

**Table 3 materials-17-02408-t003:** Experimental steel Ac3 temperatures.

Material	Empirical Formula to Calculate Ac3, °C	JMatPro Software to Calculate Ac3, °C	Experimental Test Ac3, °C
C-Mn-Mo steel	841	828	829
C-Mn-Cr steel	839	819	820

**Table 4 materials-17-02408-t004:** Mechanical properties of experimental steels at different cooling rates.

Material	Cooling Rate, (°C/s)	Yield Strength, MPa	Tensile Strength, MPa	Elongation, %	−40 °C Impact Energy, J	Hardness, HB
C-Mn-Mo steel	20	1074	1269	12.8	33	375
	25	1100	1295	12.5	34	390
	30	1121	1324	12.1	31	399
C-Mn-Cr steel	30	1075	1277	11.7	24	377
	35	1112	1310	11.3	29	398
	40	1150	1348	10.7	25	415

**Table 5 materials-17-02408-t005:** Weight loss during wear of experimental steel.

Material	Weight of Loss, mg		
	100 N, 75 rpm/min	150 N, 45 rpm/min	200 N, 20 rpm/min
C-Mn-Mo steel	38.1	73.2	156.9
C-Mn-Cr steel	30.8	79.5	177.6

## Data Availability

The raw data supporting the conclusions of this article will be made available by the authors on request.

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
