# Peer review of "Effect of Mo and Cr on the Microstructure and Properties of Low-Alloy Wear-Resistant Steels"

_materials, 2024, doi:10.3390/ma17102408_

Round 1
Reviewer 1 Report
Comments and Suggestions for Authors
Comments attached.

Author Response
Reviewer #1 comments:
Q1. The manuscript evaluated the effect of Mo and Cr on the mechanical and tribological properties of steels containing Si, Mn, Ti, Nb, P, S, and B submitted to different quenching conditions. However, the abstract and introduction section does not present a suitable context and motivation to perform the study. I disagree with the authors that there is a lack of detailed information about the effect of Mo and Cr separately on the microstructure and properties of steels. These alloying elements are some of the most studied elements in ferrous metallurgy, so true that the manuscript does not show unexpected or novel results.
Response:Thank you, we have corrected the above issue. The author has revised the abstract and introduction sections, providing appropriate background and motivation for the study. Additionally, we have made detailed revisions to inappropriate descriptions in the text, which highlights innovative results in the research. (Line 10-111)
Q2. Materials and methods should be improved significantly, especially for tribological tests. It is not clear why the author used the parameters from Table 3 where the force and speed were changed without a clear motivation. Therefore, it is not possible to define what is the crucial parameter (force or speed) in the tribological results discussion.
Response:Thanks for pointing out the mistake. The author has corrected the mistakes. The author divided the materials and methods into 2.1 materials and 2.2 processes, and corrected the problems in the research methods. Particularly, in the friction experiments, the author has provided detailed explanations of the friction experiment parameters, aiming to simulate the wear conditions during the mining and transportation processes in small to medium-sized coal mines. A comparative study of the wear performance of C-Mn-Mo steel and C-Mn-Cr steel during the wear process has been conducted. (Line 128-190)
Q3. The plots are too small concerning tables and figures, additionally, they are low-quality images.
Response:Thank you, we have corrected the above errors. The author has modified or replaced all the images in this paper and improved the resolution of the images.
Q4. The study needs to improve significantly in terms of motivation, quality, and deepness of the phenomenon investigations.
Response:Thank you. According to your opinion, we made a lot of revisions to the manuscript, fully explained the motivation and objectives of the study, and conducted a more in-depth analysis of the content of the study. At the same time, the quality of the image in the manuscript is improved.
We tried our best to improve the manuscript and we appreciate for Editors/Reviewers’ warm work earnestly, and hope that the correction will meet with approval.
Once again, thank you very much for your comments and suggestions.
Yours sincerely,
Yuxi Ma,
School of Civil Engineering and Architecture, Wuhan Polytechnic University, Wuhan, 430023, China

Reviewer 2 Report
Comments and Suggestions for Authors
The text is poorly prepared. There are many uncertainties. No detailed description of the methodology.
I. Most important comments:
1. Introduction: supplement with newer publications on this topic. Explain the importance of publication. Try to describe them separately.
2. Materials and Methods: divide into 2.1. Materials and 2.2 Process.
3. You quote standards in the text. Each should be included in the references.
4. Instead of balance, use rest (table 1).
5. If you used material markings as the starting composition, add this information. The marking of Mo steel and Cr steel is not good. There were more other ingredients. Mark as variants modified from base values.
6. In table 2, replace the slash before the units with a comma.
7. The texts in the drawings are illegible. In drawings 2 and 3, remove the units and add in mm in the drawing description. The roughness marking is no longer valid.
8. The research procedure, research process needs to be better explained.
9. Paper 22 that the authors use to determine AC3 is quite old. Are there no newer developments with improved models.
10. Describe how accurate the research equipment was. Differences in temperatures may be within the measurement error (Table 4).
11. No significant differences are visible in photos 7.
12. Figure 10 shows the tension curves. Describe the methodology and sample in Chapter 2.
13. Figures 11 and 12 show the results. Describe the methodology in chapter 2. Moreover, chapter 3.4 is too short.
II. Other notes:
14. Explain what ε-carbides means?
15. Explain the simulation conditions in JMatPro software.
16. How was the temperature accuracy shown in Figure 5 achieved?
17. Figure 4 should be improved, add more details.
III. A few minor linguistic errors (since the lines are not numbered or the pages are not numbered, these will be general comments):
-The-the,
-Wear-Wear,
-Instead of writing with lowercase letters, drawing descriptions are written with capital letters.
-Check the formatting of the text and literature.
Author Response
Response to reviewers
Dear editor and reviewers,
Thank you for offering us an opportunity to improve the quality of our submitted manuscript (Effect of Mo and Cr on the Microstructural and Properties of Low Alloy Wear-Resistant Steels, materials-2947353). We appreciated very much the reviewers’ constructive and insightful comments. In this revision, we have addressed all of these comments. We hope the revised manuscript has now met the publication standard of your journal.
We highlighted all the revisions in blue colour.
On the next pages,our point-to-point responses to the queries raised by the reviewers are listed.
Reviewer #2
- Most important comments:
Q1. Introduction: supplement with newer publications on this topic. Explain the importance of publication. Try to describe them separately.
Response:Thanks for your suggestion. The authors supplement the latest research literature, the research background and the importance of publication is described in detail again. (Line 41-111)
Q2. Materials and Methods: divide into 2.1. Materials and 2.2 Process.
Response:Thanks for your suggestion. The author provided a better explanation of the materials and methods. (Line 112-190)
Q3. You quote standards in the text. Each should be included in the references.
Response:Thank you, we very much agree with you that the author has described the criteria of the experiment in a more reasonable way so that it can be reflected in the text. (Line 154-169)
Q4. Instead of balance, use rest (table 1).
Response:The author has made modifications in Table 1. (Line 126-127)
Q5. If you used material markings as the starting composition, add this information. The marking of Mo steel and Cr steel is not good. There were more other ingredients. Mark as variants modified from base values.
Response:Thanks for your suggestion. he author modified the names of the experimental materials, changing Mo steel and Cr steel to C-Mn-Mo steel and C-Mn-Cr steel.
Q6. In table 2, replace the slash before the units with a comma.
Response:The author has made modifications in Table 2. (Line 127-128)
Q7. The texts in the drawings are illegible. In drawings 2 and 3, remove the units and add in mm in the drawing description. The roughness marking is no longer valid.
Response:Thanks for pointing out the mistake. The author has corrected the mistakes. In addition, the author has also improved the quality of the images. (Line 142-146 and Line 170-171)
Q8. The research procedure, research process needs to be better explained.
Response:The author has revised the second part of materials and methods, the third part which is the research results, the fourth part analysis and discussion, and the fifth part conclusion, making the research process clearer and delving deeper into the research results.
Q9. Paper 22 that the authors use to determine AC3 is quite old. Are there no newer developments with improved models.
Response:The author has improved the empirical formula of Ac3 based on the latest research progress and cited the latest references. (Line 201-206)
Q10. Describe how accurate the research equipment was. Differences in temperatures may be within the measurement error (Table 4).
Response:The Gleeble-3800 thermal simulation testing machine used in the study has an Ac3 temperature error range of 829 ± 1℃ and 820 ± 1 ℃ for 1 and 2, respectively. (Line 129-130)
Q11. No significant differences are visible in photos 7.
Response:We made metallographic observations with the new specimen again, and has provided an improved and enlarged image, which enhances the quality of the image. From the image, it can be seen that the hot-rolled microstructure of C-Mn Mo steel and C-Mn Cr steel are similar, both composed of ferrite and pearlite. (Line 240-244)
Q12. Figure 10 shows the tension curves. Describe the methodology and sample in Chapter 2.
Response:The author added a detailed description in 2 materials and methods. (Line 155-158)
Q13. Figures 11 and 12 show the results. Describe the methodology in chapter 2. Moreover, chapter 3.4 is too short.
Response:The author has provided detailed descriptions of the experimental methods and procedures in the second part on materials and methods. In addition, the author conducted a more detailed analysis of the experimental results in the third part and added the discussion content in the fourth part. (Line 158-171)
- Other notes:
Q14. Explain what ε-carbides means?
Response:During the tempering process, carbon in the martensite precipitates in the form of ε-carbides, reducing the supersaturation of martensite. ε-carbide is a non-equilibrium transition phase with a molecular formula of Fe2.4C. It is finely dispersed in the martensite in a needle-like manner and maintains a coherent relationship with the matrix.
Q15. Explain the simulation conditions in JMatPro software.
Response:The author inputted the chemical composition of the experimental steel in JMatPro software, the quenching temperature was set to 890℃ according to the actual production data, and the holding time to 60 minutes, and simulated the CCT curve of the experimental steel. (Line 140-142)
Q16. How was the temperature accuracy shown in Figure 5 achieved?
Response:The temperature accuracy is within 2℃. The expansion amount of experimental steel during the heating process was tested with the Gleeble-3800 thermal simulation tester, and the thermal expansion curve exhibited a turning point at the phase transition temperature. The Ac3 was determined using the tangent method, validating the accuracy of the temperature simulated in Figure 5. (Line 208-210)
Q17. Figure 4 should be improved, add more details.
Response:Thanks for your suggestion. The author added more details in Figure 4. (Line 189-190)
III. A few minor linguistic errors (since the lines are not numbered or the pages are not numbered, these will be general comments):
-The-the,
-Wear-Wear,
-Instead of writing with lowercase letters, drawing descriptions are written with capital letters.
-Check the formatting of the text and literature.
Response:Thank you, we have corrected the above errors. (Line 127-128, Line 315-316, Line 485-569)
We tried our best to improve the manuscript and we appreciate for Editors/Reviewers’ warm work earnestly, and hope that the correction will meet with approval.
Once again, thank you very much for your comments and suggestions.
Yours sincerely,
Yuxi Ma,
School of Civil Engineering and Architecture, Wuhan Polytechnic University, Wuhan, 430023, China

Reviewer 3 Report
Comments and Suggestions for Authors
1. The motivation and objective of the work should be given briefly in the abstract.
2. The quality of the figure 5 is very poor and it is difficult to find the features of the graphs. Please provide the improved and enlarged images.
3. Please providet the details of the multiple alloyying aelemetns.. why the necessity of raising the temperature.
4. The indication of the ferrite zones in figure 7 is not shhowing any clear evidence, please provide the alternative microstructures.
5. The quantity of the phases must be provided.
6. It is recommended to provide teh evidence for the increasing of cooling rate causes in increase of hardness.
7. According to figure 10, there must be a significant difference between MS and Cr steel in thier elongation, but it seems almost same for both of the steels, please verify test results to condfirm these data.
8. Does the cooling rate has an effect on the microstructure of Cr steel and phases modifications ?
Comments on the Quality of English Languageminor
Author Response
Reviewer #3 comments:
Q1. The motivation and objective of the work should be given briefly in the abstract.
Response:Thanks for your suggestion. The author has briefly explained the motivation and objective of the work in the abstract. (Line 10-16)
Q2. The quality of the figure 5 is very poor and it is difficult to find the features of the graphs. Please provide the improved and enlarged images.
Response:Thanks for pointing out the mistake. The author has corrected the mistakes and provided the improved and enlarged images. (Line 237-238)
Q3. Please provide the details of the multiple allaying elements.. why the necessity of raising the temperature.
Response:Based on the doubts, the author has consulted a large number of references and identified the latest Ac3 empirical formula. The author has made corrections to the erroneous descriptions in the text. (Line 210-214)
Q4. The indication of the ferrite zones in figure 7 is not showing any clear evidence, please provide the alternative microstructures.
Response:The author provided the clearer microstructures. (Line 243-244)
Q5. The quantity of the phases must be provided.
Response:According to the request, the author provided the quantity of phases. (Line 275-277)
Q6. It is recommended to provide the evidence for the increasing of cooling rate causes in increase of hardness.
Response:Thanks for your suggestion. The author provides evidence that an increase in cooling rate leads to an increase in hardness. (Line 245-272)
Q7. According to figure 10, there must be a significant difference between MS and Cr steel in their elongation, but it seems almost same for both of the steels, please verify test results to confirm these data.
Response:In order to verify the test results, the author redid the experiment and found that the elongation of C-Mn-Mo steel was higher than that of C-Mn-Cr steel,the difference between them was not significant, this possible factor from the martensite (Line 319-324)
Q8. Does the cooling rate has an effect on the microstructure of Cr steel and phases modifications?
Response:The effect of cooling rate on the microstructure and phases modifications of C-Mn-Cr steel. With the increase of cooling rate, the microstructure of C-Mn-Cr steel changes from Martensite + Bainite + ferrite to martensite + a little bainite, the effect of cooling rate on the microstructure evolution of experimental steel is explained in detail. (Line 256-272)
We tried our best to improve the manuscript and we appreciate for Editors/Reviewers’ warm work earnestly, and hope that the correction will meet with approval.
Once again, thank you very much for your comments and suggestions.
Yours sincerely,
Yuxi Ma,
School of Civil Engineering and Architecture, Wuhan Polytechnic University, Wuhan, 430023, China

Round 2
Reviewer 2 Report
Comments and Suggestions for Authors
- Use rpm/min units.
- It is: "Prepare Φ10 mm thick tensile specimens..." - if they were round, then the diameter, not the thickness.
- Detail markings have been removed from some of the photos - why?
- Describe in the experiment methodology where the hardness was measured (line and points).
- The text should be checked for appropriate spaces, punctuation marks, etc.
- Use the same system of symbols and units for parameters throughout the text.
-Check the format corresponding to this magazine throughout the text, e.g. is 11 ~ 13% should it be 11 - 13%?
-Check if all publications are referenced in the text? Publication 34 is quite old, please add a newer one from this scope.
Author Response
Response to reviewers
Dear editor and reviewers,
Thank you for offering us an opportunity to improve the quality of our submitted manuscript (Effect of Mo and Cr on the Microstructural and Properties of Low Alloy Wear-Resistant Steels, materials-2947353). We appreciated very much the reviewers’ constructive and insightful comments. In this revision, we have addressed all of these comments. We hope the revised manuscript has now met the publication standard of your journal.
We highlighted all the revisions in blue colour.
On the next pages,our point-to-point responses to the queries raised by the reviewers are listed.
Q1. Use rpm/min units.
Response:Thanks for your suggestion. The author has corrected the mistakes. (Line 186-187 and Line 355)
Q2. It is: "Prepare Φ10 mm thick tensile specimens..." - if they were round, then the diameter, not the thickness.
Response:Thanks for pointing out the mistake. The author has corrected the mistakes. (Line 154)
Q3. Detail markings have been removed from some of the photos - why?
Response:The author added the details in the photos. (Line 271-273)
Q4. Describe in the experiment methodology where the hardness was measured (line and points).
Response:Thanks for your suggestion. In the experimental method, the author describes in detail the location of the hardness measurement. (Line 164-170)
Q5. The text should be checked for appropriate spaces, punctuation marks, etc.
Response:Thanks for pointing out the mistake. The author has corrected the mistakes. (Line 53, Line 213, Line 234-236, Line 414-415)
Q6. Use the same system of symbols and units for parameters throughout the text.
Response:Thanks for pointing out the mistake. The author has corrected the mistakes. (Line 64, Line 126, Line 143, Line 172, Line 327-333, Line 355, Line 445)
Q7. Check the format corresponding to this magazine throughout the text, e.g. is 11 ~ 13% should it be 11 - 13%?
Response:Thanks for pointing out the mistake. The author has corrected the mistakes. (Line 126, Line 320-321)
Q8. Check if all publications are referenced in the text? Publication 34 is quite old, please add a newer one from this scope.
Response:The author revised the misquotes in the article and added newer references to replace the literature 34. (Line 336-337 and Line 555-556)
We tried our best to improve the manuscript and we appreciate for Editors/Reviewers’ warm work earnestly, and hope that the correction will meet with approval.
Once again, thank you very much for your comments and suggestions.

Reviewer 3 Report
Comments and Suggestions for Authors
The quality of the revised manuscript has been enhanced. I appreciate the author's efforts in the revision.
Comments on the Quality of English Languageminor modifications are required.
Author Response
Response to reviewers
Dear editor and reviewers,
Thank you for offering us an opportunity to improve the quality of our submitted
manuscript (Effect of Mo and Cr on the Microstructural and Properties of Low Alloy
Wear-Resistant Steels, materials-2947353). We appreciated very much the reviewers’
constructive and insightful comments. In this revision, we have addressed all of these
comments. We hope the revised manuscript has now met the publication standard of
your journal.
We highlighted all the revisions in blue colour.
On the next pages,our point-to-point responses to the queries raised by the
reviewers are listed.
Q1. Comments on the Quality of English Language minor modifications are
required.
Response: Thanks for your suggestion. The author has improved the
language in the text to ensure that they meet the quality standards of English
writing. (Line 10-12, Line 31-36, Line 40-45, Line 50, Line 76, Line 163-170, Line
234-236, Line 327-333, Line 357-368, Line 417-424, Line 453-454, Line 460-463,
Line 464-465)
We tried our best to improve the manuscript and we appreciate for
Editors/Reviewers’ warm work earnestly, and hope that the correction will meet with
approval.
Once again, thank you very much for your comments and suggestions
